# The Propensity of the Human Liver to Form Large Lipid Droplets Is Associated with PNPLA3 Polymorphism, Reduced INSIG1 and NPC1L1 Expression and Increased Fibrogenetic Capacity

**DOI:** 10.3390/ijms22116100

**Published:** 2021-06-05

**Authors:** Flaminia Ferri, Simone Carotti, Guido Carpino, Monica Mischitelli, Alfredo Cantafora, Antonio Molinaro, Maria Eva Argenziano, Simona Parisse, Alessandro Corsi, Mara Riminucci, Quirino Lai, Gianluca Mennini, Gustavo Spadetta, Francesco Pugliese, Massimo Rossi, Sergio Morini, Eugenio Gaudio, Stefano Ginanni Corradini

**Affiliations:** 1Department of Translational and Precision Medicine, Sapienza University of Rome, 00185 Rome, Italy; monica.mischitelli@uniroma1.it (M.M.); cantafora.al@tin.it (A.C.); mariaeva.argenziano@gmail.com (M.E.A.); simona.parisse@uniroma1.it (S.P.); stefano.corradini@uniroma1.it (S.G.C.); 2Laboratory of Microscopic and Ultrastructural Anatomy, CIR, University Campus Bio-Medico, 00128 Rome, Italy; s.carotti@unicampus.it (S.C.); s.morini@unicampus.it (S.M.); 3Department of Movement, Human and Health Sciences, Division of Health Sciences, University of Rome “Foro Italico”, 00135 Rome, Italy; guido.carpino@uniroma4.it; 4Wallenberg Laboratory, Department of Molecular and Clinical Medicine/Wallenberg Laboratory, Institute of Medicine, University of Gothenburg and Sahlgrenska University Hospital, 413 45 Gothenburg, Sweden; antonio.molinaro@wlab.gu.se; 5Department of Molecular Medicine, Sapienza University of Rome, 00161 Rome, Italy; alessandro.corsi@uniroma1.it (A.C.); mara.riminucci@uniroma1.it (M.R.); 6General Surgery and Organ Transplantation Unit, Department of Surgery, Sapienza University of Rome, 00161 Rome, Italy; quirino.lai@uniroma1.it (Q.L.); gianluca.mennini@uniroma1.it (G.M.); massimo.rossi@uniroma1.it (M.R.); 7Department of Internal Anesthesiological and Cardiovascular Clinical Sciences, Sapienza University of Rome, 00185 Rome, Italy; gustavo.spadetta@uniroma1.it (G.S.); f.pugliese@uniroma1.it (F.P.); 8Department of Anatomical, Histological, Forensic Medicine and Orthopedics Sciences, Sapienza University of Rome, 00161 Rome, Italy; eugenio.gaudio@uniroma1.it

**Keywords:** cholesterol, liver donor, hepatic stellate cells, fibrosis, large droplet macrovesicular steatosis, lipid droplets, INSIG-1, NAFLD, NPC1L1, PNPLA3

## Abstract

In nonalcoholic steatohepatitis animal models, an increased lipid droplet size in hepatocytes is associated with fibrogenesis. Hepatocytes with large droplet (Ld-MaS) or small droplet (Sd-MaS) macrovesicular steatosis may coexist in the human liver, but the factors associated with the predominance of one type over the other, including hepatic fibrogenic capacity, are unknown. In pre-ischemic liver biopsies from 225 consecutive liver transplant donors, we retrospectively counted hepatocytes with Ld-MaS and Sd-MaS and defined the predominant type of steatosis as involving ≥50% of steatotic hepatocytes. We analyzed a donor Patatin-like phospholipase domain-containing protein 3 (PNPLA3) rs738409 polymorphism, hepatic expression of proteins involved in lipid metabolism by RT-PCR, hepatic stellate cell (HSC) activation by α-SMA immunohistochemistry and, one year after transplantation, histological progression of fibrosis due to Hepatitis C Virus (HCV) recurrence. Seventy-four livers had no steatosis, and there were 98 and 53 with predominant Ld-MaS and Sd-MaS, respectively. In linear regression models, adjusted for many donor variables, the percentage of steatotic hepatocytes affected by Ld-MaS was inversely associated with hepatic expression of Insulin Induced Gene 1 (INSIG-1) and Niemann-Pick C1-Like 1 gene (NPC1L1) and directly with donor PNPLA3 variant M, HSC activation and progression of post-transplant fibrosis. In humans, Ld-MaS formation by hepatocytes is associated with abnormal PNPLA3-mediated lipolysis, downregulation of both the intracellular cholesterol sensor and cholesterol reabsorption from bile and increased hepatic fibrogenesis.

## 1. Introduction

Nonalcoholic fatty liver disease (NAFLD) has an overall global prevalence of about 25.2% and is expected to increase in the near future [1,2]. The disease, characterized by hepatic steatosis, may proceed to nonalcoholic steatohepatitis (NASH) and, eventually, fibrosis with cirrhosis and/or hepatocellular carcinoma [2].

According to the Brunt [3] classification, there are three types of steatotic hepatocytes based on the size of the lipid droplets (LDs) that accumulate in the cytoplasm and the position of the nucleus: (a) large droplet macrovesicular steatosis (Ld-MaS), described as a single large lipid droplet which displaces the nucleus; (b) small droplet macrovesicular steatosis (Sd-MaS), characterized by the presence of multiple lipid droplets without nucleus displacement; and (c) true microvesicular steatosis, when the cytoplasm acquires a foamy appearance due to the formation of miniscule undiscernible lipid droplets that are diffusely distributed without nucleus displacement [4]. In the past literature, Sd-MaS has been labeled as microvesicular steatosis [5]. However, in Sd-MaS, the fatty vacuoles are larger and do not show a diffuse distribution pattern as in true microvesicular steatosis [4].

Several experimental data have linked a greater size of LDs in hepatocytes and an increase in their size over time to the severity of liver fibrosis in NAFLD models [6,7,8,9,10]. In humans, hepatic fibrosis in NAFLD/NASH has been associated with the severity of macrovesicular steatosis [11] and the presence of true microvesicular steatosis [12]. There is a lack of studies linking liver fibrosis to the different types of macrovesicular steatosis according to the size of LDs and the Brunt classification [3].

Cholesterol is rapidly emerging as a key metabolite in the pathogenesis and severity of NAFLD [13,14]. In experimental models of NAFLD, cholesterol is more associated with large than small LDs in hepatocytes [15,16]. The size of LDs after their formation depends on very complex mechanisms including their catabolism through lipolysis and different types of lipophagy, their eventual fusion and, probably, also lipid synthesis on their surface [10,17,18,19,20]. Patatin-like phospholipase domain protein 3 (PNPLA3) is one lipolytic enzyme in hepatocytes located on large lipid droplets and several studies have shown that the PNPLA3 rs738409 (I148M) variant is powerfully linked to hepatic damage in NAFLD, at different levels such as steatosis, necroinflammation and fibrosis [21,22]. Although in vitro and animal studies have shown that the PNPLA3 M variant is associated with accumulation of larger LDs in hepatocytes, no data are available in humans [23,24,25,26].

The percentage of steatotic hepatocytes in the liver and the size of LDs in each steatotic hepatocyte are determined not only by the imbalance between lipid supply and utilization/secretion but also by the amount/activity of regulatory surface proteins, including PNPLA3, and of lysosomal enzymes, which can differ in the hepatocytes of the same liver [17,20,21,27,28,29,30]. As a result, in the same human liver, heterogeneous populations of hepatocytes with and without steatosis can coexist, and, among the former, sometimes the size of the LDs is uniform, or some hepatocytes may contain large and others small LDs.

Dysregulation of lipid metabolism due to the M variant of PNPLA3 could favor the development of NAFLD-associated disease, such as cardiovascular disease (CVD). In fact, since NAFLD occurs in the context of systemic metabolic dysfunction, the risk of CVD and diabetes is spontaneously intensified.

Whether the same heritability component that affects NAFLD could also play a role in CVD development in NAFLD patients is not clear.

The scientific literature is controversial: as evidenced by Nobili et al. [31], nutritional genomics or the impact of diet on disease development by modulating the expression of an individual’s genetic pattern suggests that carbohydrate and sugar intake is related to hepatic triglyceride accumulation exclusively in homozygous PNPLA3 M carriers. Probably, carbohydrates upregulate PNPLA3 expression, which in turn favors the accumulation of the pathological protein on the surface of lipid droplets. Since it has been demonstrated that excessive sugar consumption exerts a negative outcome on CVD risk [31], PNPLA3 unfavorable genetics could amplify this effect.

On the contrary, the association of the unfavorable PNPLA3 allele with CVD risk lacked in a Chinese study considering a cohort of 189 patients with NAFLD and CVD and 242 patients with NAFLD.

Serum levels of triglycerides and low-density lipoprotein (LDL) resulted in being significantly lower in PNPLA3 M carriers with respect to wild-type individuals, suggesting a protective effect of the mutated allele [32].

An interesting study in which 157 Caucasian patients were enrolled with type 2 diabetes evidenced the role of the PNPLA3 M variant in increasing the risk of chronic kidney disease (CKD). In particular, in 112 NAFLD patients and 43 CKD subjects, it was shown that the estimated glomerular filtration rate was lower in the carriers of the variant than in wild-type people [33].

In conclusion, whether or not PNPLA3 rs738409 could play a significant role in other diseases beyond NAFLD is still unclear. More correctly, we should consider the pleiotropic effect of this gene and its variants rather than its single impact on disease development.

For this reason, PNPLA3 is a good-looking target for therapeutic intervention aimed to avoid the progression of fatty liver disease towards the more critical stages.

Momelotinib, a drug evaluated for myelofibrosis therapeutics, revealed its strong potential as an inhibitor of PNPLA3 expression in a dose-dependent manner both in hepatocytes and stellate cells [34]. Therefore, it could be a candidate to block the profibrogenic and proinflammatory effects of PNPLA3 I148M.

Additionally, silencing of a PNPLA3-mutated gene turned out as a good strategy to improve nonalcoholic steatohepatitis and fibrosis. The administration of an antisense oligonucleotide that mediated the silencing of the PNPLA3 gene demonstrated, in 148M/M mice, a reduction in liver steatosis although the mice were fed with a steatogenic high-sucrose diet [35].

Hence, PNPLA3 gene interference should be taken in great consideration to develop a new strategy that disturbs this disease-causing gene.

In the present study, we decided to use a novel approach to analyze factors associated with steatosis based on the propensity of the liver to produce relatively more Ld-MaS or predominantly Sd-MaS. To do this, we calculated the percentage of the two types of steatosis with respect to the total number of steatotic hepatocytes. In this way, we assessed the percentage of steatotic hepatocytes affected by Ld-MaS, and therefore its inverse, that is, the percentage affected by Sd-MaS, regardless of what the percentage of each of the two types of steatosis was with respect to the total number of hepatocytes, as was usually conducted in the past.

We performed routine liver biopsies in donors at the time of liver transplantation (LT), who, as known, have a high prevalence of steatosis [5], to investigate which factors immediately prior to LT were associated with the propensity of the liver to accumulate one type of steatosis rather than another, according to the Brunt classification [3], including the presence in the donor of the PNPLA3 rs738409 (I148M) variant, hepatic expression at the mRNA level of PNPLA3 and some proteins involved in cholesterol metabolism and the degree of activation of hepatic stellate cells (HSCs), the cells responsible for fibrogenesis. Furthermore, we investigated whether the propensity of the liver to accumulate one type of steatosis rather than another was related to subsequent graft fibrosis in post-transplant biopsies performed for hepatitis c virus (HCV) recurrence.

We found that the percentage of steatotic hepatocytes affected by Ld-MaS was, independently from confounding factors of the donor, directly associated with the PNPLA3 polymorphism and adrenaline administration in the donor, HSC activation before transplantation and subsequent progression of post-transplant fibrosis. Furthermore, an independent inverse association was found between the percentage of steatotic hepatocytes affected by Ld-MaS and hepatic expression of Insulin Induced Gene 1 (INSIG-1) and **Niemann**-Pick C1-Like 1 gene (NPC1L1).

## 2. Results

### 2.1. Characteristics of Donors and Percentage Distribution of the Types of Liver Graft Steatosis with Respect to the Total Number of Hepatocytes in the Entire Study Population

Donors were predominantly males, with a median age of 51 years, a median BMI of 24.8 kg/m^2^ and a median ICU (intensive care unit) stay of 3 days, where one third had a traumatic cause of death and half were treated with noradrenaline infusion (Appendix A).

An excellent inter-analytical correlation (interclass correlation coefficient >0.9) was reported between the two histopathologists concerning steatosis.

The distribution of Ld-MaS or Sd-MaS, expressed as the respective percentage in the entire hepatocyte population, was as follows: liver graft Ld-MaS and Sd-MaS involving <5% of hepatocytes was present in 48.9% and in 68% of cases, respectively. Ld-MaS >33% was present in only 4.4% of grafts, with a maximum Ld-MaS value of 40% observed in six cases. Sd-MaS >33% was present in 7.1% of cases, with nine grafts having >40% of Sd-Mas and a maximum Sd-MaS value of 80% observed in one graft (Appendix A). Among the steatotic grafts, 56 (37.0%) had only Ld-MaS, 16 (10.6%) had only Sd-MaS and 79 (52.3%) had both Ld-MaS and Sd-MaS. These data show that, in the liver donor population, and considering the classical classification in which the percentage of hepatocytes affected by steatosis is calculated with respect to the total of hepatocytes, about half have less than 5% of Ld-MaS and can therefore be considered without steatosis. Furthermore, there were no grafts with Ld-MaS >40% for a selection bias as these are not used for transplantation. Finally, more than half of the grafts affected by steatosis had both hepatocytes with LD-MaS and with Sd-MaS (Figure 1).

### 2.2. Percentage Distribution of Steatosis according to the Predominant Type, Ld-Mas or Sd-MaS, Calculated on the Total of Steatotic Hepatocytes

To subgroup the steatotic liver grafts according to the predominant type, Ld-Mas or Sd-Mas, we calculated the percentage of hepatocytes affected by Ld-MaS with respect to all steatotic hepatocytes. Predominant Ld-MaS, i.e., with at least 50% of steatotic hepatocytes affected by LD-MaS, was found in 98 of 225 (43.6%). Predominant Sd-MaS, i.e., with less than 50% of steatotic hepatocytes affected by LD-MaS, was found in 53 of 225 (23.6%) grafts. As expected, grafts with predominant Ld-MaS had a significantly higher percentage of hepatocytes affected by Ld-MaS, even when calculated on the total number of hepatocytes, whereas grafts with predominantly Sd-MaS had a significantly higher percentage of hepatocytes affected by Sd-MaS, even when calculated on the total of hepatocytes (Table 1). Using the <5% cut-off of both Ld-MaS and Sd-MaSA, a total of 74 liver grafts (32.8%) had nil-Steatosis.

#### 2.2.1. Univariate Analyses of Donor Variables Associated with the Absence of Steatosis or with the Predominant Type of Steatosis

Table 1 also shows the univariate analyses of donor demographics and clinical and genetic characteristics comparing grafts with predominant Ld-MaS, those with predominant Sd-MaS and those with nil-Steatosis. No intergroup difference was found in serum AST and ALT levels. The distribution of the M allele of the PNPLA3 variant did not differ. However, there was a trend for a higher frequency of allele M homozygosis in the predominant Ld-MaS group. Liver grafts with nil-Steatosis compared to those with predominant Ld-MaS had a significantly younger donor age, lower BMI and higher frequency of traumatic cause of death. Liver grafts with nil-Steatosis compared to those with predominant Sd-MaS had a significantly lower BMI and longer length of ICU stay, with no difference in the other variables, including PNPLA3 variant distribution. Liver grafts with predominant Ld-MaS compared to those with predominant Sd-MaS had a significantly longer ICU stay and higher frequency of noradrenaline infusion.

#### 2.2.2. Univariate Analyses of Gene Expression of Hepatic Graft Lipid Metabolism and Hepatic Stellate Activation Associated with the Presence and the Predominant Type of Steatosis

We then investigated liver graft mRNA gene expression of PNPLA3 and APOB, involved in Ld-MaS and VLDL metabolism, and of genes involved in cholesterol hepatic metabolism according to the presence and the predominant type of steatosis in the pre-ischemia biopsy before LT. At univariate analyses (Figure 2), NPC1L1 mRNA expression was significantly lower in liver grafts with predominant Ld-MaS compared to both those with predominant Sd-MaS (*p* = 0.011) and those with nil-Steatosis (*p* = 0.003). Although for SREBP-2 and LDL-R there was a significant overall difference between the three groups, at the post hoc analysis, there were no significant intergroup differences (Appendix A). There was no correlation between PNPLA3 mRNA expression and the PNPLA3 rs738409 polymorphism in the entire study population or when investigated separately in livers with predominant Ld-MaS, predominant Sd-MaS or nil-Steatosis. We then investigated liver graft total lobular and zonal activation of HSCs according to the presence and the predominant type of steatosis in the pre-ischemia biopsy before LT. At univariate analyses, liver grafts with predominant Ld-MaS compared to those with predominant Sd-MaS had significantly higher total lobular and acinar zones 2 and 3 HSC activation (Table 2 and Figure 3). No intergroup difference was found when comparing liver grafts without steatosis with either those with predominant Ld-MaS or predominant Sd-MaS.

#### 2.2.3. Multivariate Analyses of Factors Associated with Each Predominant Type of Steatosis vs. Nil-Steatosis

We performed multivariate binary regression models to discover factors before LT independently associated with each predominant type of steatosis vs. nil-Steatosis. Predominant Ld-MaS compared to nil-Steatosis was significantly associated with donor age (O.R. 1.028 [CI 1.006–1.050]; *p* = 0.012) and donor BMI (O.R. 1.247 [CI 1.089–1.427]; *p* = 0.001). Regarding the donor PNPLA3 polymorphism, although the M allele was not associated with predominant Ld-MaS vs. nil-Steatosis, homozygosity (MM) compared to the wild type (II) was significantly associated with predominant Ld-MaS compared to nil-Steatosis in a dominant model (O.R. 4.261 [CI 1.017–17.845]; *p* = 0.047). With regard to the mRNA gene expression of lipid metabolism, predominant Ld-MaS compared to nil-Steatosis was significantly associated with low liver graft NPC1L1 (O.R. 0.137 [CI 0.031–0.603]; *p* = 0.009), SREBP-2 (O.R. 0.762 [CI 0.600–0.968]; *p* = 0.026) and PNPLA3 (O.R. 0.916 [CI 0.844–0.995]; *p* = 0.038) mRNA expression (Figure 4). Predominant Sd-MaS compared to nil-Steatosis was significantly associated with donor BMI (O.R. 1.218 [CI 1.041–1.425]; *p* = 0.014) and length of ICU stay (O.R. 0.833 [CI 0.726–0.956]; *p* = 0.009). These data show that livers with predominant Ld-MaS compared to those without steatosis are independently associated with: (a) a more advanced age; (b) higher BMI; (c) being homozygous for the PNPLA3 polymorphism and lower expression of the protein encoded by the PNPLA3 gene, signs of abnormal lipolysis; and (d) lower expression of SREBP-2 and NPC1L1, indicating reduced activation of the cholesterol reabsorption pathway from bile. Regarding the comparison between livers without steatosis and those with predominant Sd-MaS, the latter are independently associated with: (a) a higher BMI; and (b) a shorter ICU stay, probably the expression of a shorter duration of undernutrition.

#### 2.2.4. Linear Regression Analyses of the Factors Associated with the Percentage of Steatotic Hepatocytes Affected by Ld-MaS

Table 3 shows the linear regression analyses of the variables before LT which we found to be significantly associated with the percentage of steatotic hepatocytes affected by Ld-MaS. After adjustment for confounders, we found that the percentage of steatotic hepatocytes affected by Ld-MaS was significantly associated with the donor PNPLA3 variant M allele and noradrenaline administration, low liver graft INSIG1 and NPC1L1 mRNA expression and high total and zonal HSC activation (Figure 2). These data show that, among livers with steatosis, a predisposition to form Ld-MaS, rather than Sd-MaS, is independently associated with: (a) the M variant of PNPLA3, a sign of abnormal lipolysis; (b) the administration of noradrenaline, and therefore poor blood perfusion of the liver; (c) a low hepatic expression of INSIG1 and NPC1L1, indicating sensing of a high intracellular cholesterol content and reduced activation of the cholesterol reabsorption pathway from bile; and (d) a high rate of HSC activation, a sign of predisposition to fibrogenesis.

#### 2.2.5. Association between the Percentage of Steatotic Hepatocytes Affected by Ld-MaS before Transplant and the Progression of Fibrosis due to Recurrent HCV Disease

Thirty-four liver grafts that were steatotic at the biopsy before transplantation underwent a biopsy for HCV recurrence after the LT at a median time of 13.25 (CI 9.00–21.25) months after the operation. We investigated whether the predisposition of the hepatic graft to form Ld-MaS, measured immediately before transplantation, was correlated with the subsequent progression of hepatic fibrosis in the biopsy performed after the transplant for HCV recurrence. At linear regression analysis, there was a positive correlation between the percentage of steatotic hepatocytes affected by Ld-MaS in the pre-transplant biopsy and the progression of fibrosis at post-transplant biopsy (t = 2.821; Beta 0.002 [CI 0.001–0.004]; *p* = 0.008). The linear association remained significant also after adjustment for serum HCV quantitative RNA at the time of re-biopsy (t = 2.193; Beta 0.002 [CI 0.000–0.003]; *p* = 0.037). These data demonstrate that, among livers with fat accumulation in the hepatocytes, a predisposition to form Ld-MaS, rather than Sd-MaS, independently associates with the subsequent fibrogenic capacity of the liver.

## 3. Discussion

In the present study, we investigated which donor and liver graft factors, at the time of LT, are associated with organ predisposition to form one type or another of steatosis according to the size of macrovesicular LDs, large or small. These factors included the donor PNPLA3 rs738409 polymorphism genotype as well as liver expression of genes involved in lipid metabolism and HSC activation. Our investigation also considered the association of the prevalent type of steatosis with subsequent liver graft fibrosis, in subjects with recurrence of HCV.

The main associations, independent of confounding factors, that we found are as follows: (a) the percentage of steatotic hepatocytes affected by Ld-MaS in the liver graft is associated with the donor PNPLA3 rs738409 M allele variant and noradrenaline administration, higher HSC activation in zones 2 and 3 of the hepatic lobule and lower hepatic INSIG-1 and NPC1L1 mRNA expression before LT and the rate of the progression of fibrosis for HCV recurrence after LT; (b) predominant Ld-MaS compared to nil-Steatosis of the liver graft is associated with donor homozygosity for the M allele of the PNPLA3 variant, increased age and BMI and lower hepatic mRNA expression of PNPLA3 and NPC1L1; (c) predominant Sd-MaS compared to nil-Steatosis of the liver graft is associated with an increased BMI and shorter length of ICU stay.

Here, for the first time in humans, we report that the M allele of the PNPLA3 rs738409 variant coding for a substitution from isoleucine to methionine at position 148 is associated with the ability of the liver to develop Ld-MaS in the majority of steatotic hepatocytes, compared to those that mainly develop Sd-MaS. We also found an association of the donor homozygosity for the M allele of PNPLA3 with predominant Ld-MaS when the latter was compared with nil-Steatosis in the liver.

These data are in line with previous experimental studies showing a larger LD size in models of the PNPLA3 variant and in a knock-in mouse model homozygous for the PNPLA3 M allele [23,24,25,26]. We also found that a low hepatic expression of PNPLA3 mRNA was independently associated with predominant Ld-MaS vs. nil-Steatosis. PNPLA3 mRNA was not related to the PNPLA3 rs738409 variant distribution, suggesting an epigenetic reduction in the PNPLA3 function in livers predisposed to form large LDs, in agreement with a recent report in NAFLD patients [36].

The second finding of the present study was a strong association between the propensity of hepatocytes to accumulate Ld-MaS, on the one hand, and both activation of HSC at the same time and subsequent progression of fibrosis during HCV infection, on the other. This is consistent with several experimental data. In fact, activation of HSC and fibrosis occurs only in some rat strains under a high-fat diet which develop steatosis with large lipid droplets, but not in those that accumulate small lipid droplets [6,7]. Furthermore, in animal models of NASH, a parallel time-dependent increase in LD size in hepatocytes and deposition of hepatic fibrosis has been demonstrated by automated computerized image analysis [8]. Finally, a parallel reduction in lipid droplet size and fibrosis was described in a liver-specific knockout mouse model for perilipin 2, an LD coating protein that inhibits lipolysis and regulates LD size, under a steatogenic diet [9,10].

Regarding human studies, in NAFLD/NASH, hepatic fibrosis has been associated with true microvesicular steatosis [12], the prevalence of which in our present study, as in another population of transplant donors, does not exceed 1%, probably due to the selection process of donor candidates [4].

Our study is the first to demonstrate the association between the propensity of hepatocytes to accumulate large rather than small LDs and concomitant activation of HSC in humans. Our finding that the progression of fibrosis in the clinical model of HCV recurrence after LT is more rapid in livers that had a predominant Ld-MaS before LT is in agreement with what has already been reported in the past for the severity of macrovesicular steatosis [37,38]. Large LDs in hepatocytes could directly cause fibrosis through increased compression of the microcirculation and impaired balance of vasodilators and sinusoidal constrictors, leading to activation of HSCs [6]. Alternatively, larger LDs may not directly be the cause of fibrosis progression, but simply the expression of an altered function of their surface proteins which is the true cause of the fibrosis. Indeed, the knockdown of perilipin 2 protects from endoplasmic reticulum stress and inflammation secondary to a high-fat diet, and a genetically determined abnormal function of PNPLA3 can directly activate HSCs [9,21,22]. In our present study, however, we did not find any correlation between the PNPLA3 polymorphism and HSC activation (data not shown).

The third factor that we found associated with liver grafts characterized by predominant Ld-MaS, compared to those with predominant Sd-MaS, is the administration of noradrenaline to the donor. Liver grafts harvested from donors who undergo noradrenaline administration are likely to have suffered hypoxia. Noradrenaline reduces oxygen uptake, triglyceride secretion and intermittent hypoxia, such as in patients with obstructive sleep apnea, and promotes steatosis and NASH [39,40]. Hypoxia could lead to an increased size of LDs by inducing hepatic hypoxia-inducible factor 1α and 2α and/or Hypoxia Inducible Lipid Droplet Associated protein expression [41,42,43].

Since cholesterol accumulation in the liver is involved in NAFLD pathogenesis and severity, we investigated hepatic mRNA expression of key proteins involved in cholesterol metabolism, in relation to lipid droplet size [13]. We found that the percentage of steatotic hepatocytes affected by Ld-MaS correlated negatively with hepatic expression of NPC1L1 and INSIG-1. NPC1L1 in the liver promotes intrahepatic reabsorption of free cholesterol from bile into hepatocytes [44]. INSIG-1 is a sensor of intracellular cholesterol levels, and its low expression indicates a high cholesterol content [45]. Therefore, although we did not measure hepatocellular cholesterol content in the present study, we hypothesize that the low expression of NPC1L1 in hepatocytes with predominant Ld-MaS represents a feedback downregulation due to the high free cholesterol content. Consistent with our hypothesis, the large lipid droplets in hepatocytes contain more cholesterol than the small lipid droplets in a mouse model of NASH [15], and the size of the lipid droplets increases over time in a cholesterol-fed rat model of NASH [16]. A recent meta-analysis showed that administration of ezetimibe, an NPC1L1 inhibitor, improves liver inflammatory activity but not steatosis in patients with NASH [46]. This last finding could be due to the fact that livers with predominant Ld-MaS, which in our study are the majority, have a very low expression of NPC1L1, lower than both that of livers with predominant Sd-MaS and that of livers without steatosis. Our data cannot explain the reciprocal relationships we found between accumulation of Ld-MaS in hepatocytes, low expression of NPC1L1 and INSIG-1 in the liver and high activation of HSCs. We can hypothesize that in livers with predominant Ld-MaS, in which the low expression of NPC1L1 and INSIG-1 reflects a high content of free cholesterol, like the hepatocytes, HSC may also have an increased content of free cholesterol and that the latter, as demonstrated in animal models, causes a greater activation of HSCs [47].

In our study, we found that predominant Ld-MaS compared to nil-Steatosis was associated with older donor age. This is consistent with the experimental demonstration that hepatocyte senescence causes reduced mitochondrial ᵦ-oxidation and consequent accumulation of predominantly large lipid droplets [17,48].

Finally, predominant Sd-MaS was associated with a shorter stay in the ICU. As the ICU stay and reduced caloric intake prolong, Sd-MaS is likely to be reduced by lipophagy activation [18] and by fusion of small LDs induced by activation of cell death-inducing DNA fragmentation factor-α-like effector proteins on their surface [27].

This study has some limitations. It is a monocenter study, and our data on the predominant type of steatosis were obtained in a selected population such as cadaveric liver donors utilized for LT. This implies that livers with fibrosis were excluded as well as those with more than 40% of Ld-MaS assessed as the percentage of the entire hepatocytes (steatotic plus non-steatotic). Furthermore, although we adjusted our analyses for some confounding factors such as the cause of brain death, length of ICU stay and noradrenaline administration, donors do not exactly represent NAFLD patients of the general population, due to brain death, because they undergo prolonged fasting and are treated aggressively in the ICU. However, we believe that our results on the PNPLA3 variant, on the low expression of INSIG-1 and NPC1L1 representing a high intracellular cholesterol content and on the activation of HSC and fibrosis as factors associated with a particular attitude of the liver for the formation of Ld-MaS can be extrapolated to NAFLD patients.

## 4. Materials and Methods

### 4.1. Liver Donors

We retrospectively analyzed 256 hepatic grafts of cadaveric heart beating donors submitted to protocol pre-ischemia liver biopsies, enrolled consecutively in the LT Center of the Sapienza University of Rome (Italy) from February 2001 to October 2013. The study followed the ethical procedures approved by the ethical committee of Policlinico Umberto I and Sapienza University of Rome. As shown in Figure 5, exclusion criteria were defined as the exclusive presence of true microvesicular steatosis (*n* = 2), biopsy inadequate for steatosis evaluation (*n* = 27) or the presence of fibrosis (*n* = 2). The study was conducted on the remaining 225 liver graft donors. In addition to liver graft retrospective re-estimation according to the Brunt classification [3], anthropometric and clinical donor data from clinical records were prospectively collected from the National Transplant Center database. Among the 225 donors, PNPLA3 genotyping was successfully performed in 152 cases and, on the liver grafts of the remaining donors, we evaluated the expression of genes involved in hepatic lipid metabolism in 97 cases. Finally, among the latter, we evaluated, by immunohistochemistry, HSC activation in 63 grafts.

### 4.2. Liver Graft Biopsies at Transplant

Protocol wedge liver biopsies were performed in all donors, immediately before aorta clamping from the lower rear margin of the left hepatic lobe. One half of each biopsy was immediately fixed in 10% neutral buffered formalin, and then we created paraffin sections successively stained with hematoxylin and eosin or used for immunohistochemistry. The other half was immediately snap frozen in liquid nitrogen and used to quantify the expression of selected genes.

### 4.3. Liver Graft Steatosis Assessment

Permanent histological sections were analyzed to assess and classify the donor liver steatosis according to the Brunt classification [3,4,48]: Ld-MaS, as one or few large vacuoles in the cytoplasm with nuclear displacement in an eccentric location of the cell; Sd-MaS, as few and discrete fat vacuoles that were smaller than half of the cell and did not displace the nucleus (Figure 2). True microvesicular steatosis was defined as the presence of innumerable tiny indiscernible lipid vesicles diffusely distributed in the cytoplasm causing its foamy appearance as previously illustrated also by Ferri et al. [49]. The extent of each type of steatosis was expressed as a percentage of each type of steatotic hepatocyte compared to all hepatocytes. All evaluations were performed independently by two pathologists (AC and MR) specialized in liver pathology.

### 4.4. Liver Graft Hepatic Stellate Cell Activation

The activation of HSCs was assessed by α-Smooth muscle actin (α-SMA) immunohistochemistry using the method previously described by Carotti et al. [50]. Mouse monoclonal anti–α-SMA antibody (Dako 1A4) diluted 1:40 was used as primary antibody. Only parenchymal HSCs were considered and individuated by morphological criteria (perisinusoidally located, stellate-shaped cells residing in the parenchymal lobules or nodules). Estimation of the number of anti-α-SMA immunoreactive HSCs was performed independently by three pathologists (SC, GC and SM) specialized in liver pathology. Intraobserver agreement was higher than 90%. The number of positive HSCs and negative HSCs was counted under a light microscope at 200× magnification in the entire lobule and in acinar zones 1, 2 and 3: only the cells that displayed nuclei on the section were considered. For each slide, at least 7–10 microscopic fields were randomly chosen, and the number of HSCs was expressed as mean number per field (0.1 mm^2^) (Figure 3).

### 4.5. Liver Graft Lipid Metabolism mRNA Gene Expression

Retro transcriptional quantitative polymerase chain reaction (RT-QPCR) was performed to quantify the expression of selected genes of interest: PNPLA3, apolipoprotein B (APOB), insulin-induced gene 1 (INSIG-1), sterol regulatory element-binding protein 2 (SREBP-2), 3-hydroxyl-3-methylglutaryl coenzyme A reductase (HMGCR), NPC1L1, LDL receptor (LDLR), proprotein convertase subtilisin/kexin type 9 (PCSK9), Liver X receptor (LXR). Twenty-five mg of each liver sample was homogenized and incubated in lysis buffer and proteinase K (200 mg/mL). RNA was extracted using TRIzol reagent (Invitrogen). cDNA was obtained using High-Capacity RNA-to-cDNA Products (Invitrogen). Two micrograms of total purified cDNA were used for Quantitative PCR to quantify the genes of interest by using the 2^-ΔΔCT^ method [51]. Normalization of gene expression was performed using glyceraldehyde-3-phosphate dehydrogenase (GAPDH) as a reference gene.

### 4.6. PNPLA3 rs738409 Single-Nucleotide Polymorphism Donor Screening

DNA extraction from blood was performed by the DNeasy^®^ Tissue Kit (QIAGEN S.p.A, Milan, Italy) according to the manufacturer’s instructions. Purified genomic DNA was used using a dedicated TaqMan^®^ genotyping assay that allows distinguishing both wild-type and variant alleles at the SNP site in the DNA target sequence (TaqMan Applied Biosystems, Foster City, CA, USA). Only for 152 out of 161 samples, PCR allelic discrimination, performed twice in two independent analyses, gave a 100% concordance rate.

### 4.7. Fibrosis Progression after Transplant

To verify if the predisposition of the liver graft to form Ld-MaS, measured immediately before the transplant, was correlated with the progression of hepatic fibrosis of the same liver after the transplant, we took advantage of the biopsies performed when clinically indicated in the recipients with HCV recurrence. The immunosuppressive protocol was based on a triple therapy with methylprednisolone, mycophenolate mofetil and calcineurin inhibitor. Methylprednisolone was rapidly tapered. The progression rate of post-LT hepatic fibrosis was measured by dividing the stage of fibrosis according to Ishak et al. [52] by the number of months from the date of the transplant to that of the biopsy and was expressed in fibrosis units per month.

### 4.8. Statistical Analyses

Liver grafts were divided into three groups according to the presence and to the predominant type of steatosis at pre-ischemia biopsy in the donor. Liver grafts with less than 5% of hepatocytes affected by steatosis were considered as the “nil-Steatosis” group. Since the remaining grafts were affected by pure Ld-MaS, pure Sd-MaS or by both types of steatosis, we divided them into two groups on the basis of the predominant type of steatosis (i.e., relatively more steatotic hepatocytes containing Ld-MaS or Sd-MaS). They were classified as being affected by predominant Ld-MaS or predominant Sd-MaS by calculating the percentage of hepatocytes affected by Ld-MaS within the population of all the steatotic hepatocytes (the sum of hepatocytes affected by Ld-MaS and those affected by Sd-MaS). Livers that were affected by Ld-MaS in ≥50% of steatotic hepatocytes, including those with 100% of steatotic hepatocytes affected by Ld-MaS, were classified as “predominant Ld-MaS”. Livers that were affected by Sd-MaS in ≥50% of steatotic hepatocytes, including those with 100% of steatotic hepatocytes affected by Sd-MaS, were classified as “predominant Sd-MaS”.

For continuous variables, the normality was assessed by the Kolmogorov–Smirnov test with Lilliefors correction, and data are expressed as median and interquartile range or mean ± SE. For categorical variables, data are expressed as absolute values and percentages. To test the overall differences among the 3 groups (nil-Steatosis, predominant Ld-MaS and predominant Sd-MaS), continuous variables were analyzed with the Kruskal–Wallis test or the ANOVA one-way test, as appropriate. Post hoc analyses were performed using the Dunn test with Bonferroni correction for multiple comparisons or the Bonferroni test. For categorical variables, intergroup differences were analyzed using the χ^2^ or Fisher’s exact test, as appropriate.

Separate binary logistic regression models were used to analyze factors associated with predominant Ld-MaS or predominant Sd-MaS vs. nil-Steatosis. In the context of steatotic livers, the relationship of various factors associated with the predominant type of steatosis was investigated using crude and then multiple adjusted linear regression models. In the latter, the percentage of steatotic hepatocytes affected by Ld-MaS was entered as a dependent variable, and separate models were constructed for the donor PNPLA3 polymorphism, for the hepatic expression of lipid metabolism genes of interest and for the activation of total and zonal HSCs. Each multivariate model was adjusted for the demographic and clinical variables of the donor showing a *p*-value of <0.100 on univariate analysis. A *p*-value <0.05 was considered significant. Computations were carried out with SPSS software 25.0 for Windows (SPSS Inc., Chicago, IL, USA).

## 5. Conclusions

In summary, our study demonstrates that the propensity of human hepatocytes to accumulate in vivo large lipid droplets is associated with the PNPLA3 polymorphism, low hepatic PNPLA3, INSIG-1 and NPC1L1 transcript abundance, probably signs of high hepatocellular cholesterol content and reduced cholesterol reabsorption from bile, HSC activation and subsequent fibrosis progression. Our findings are clinically relevant because they link a particular type of steatosis to a more severe evolution of chronic liver disease and, from a precision medicine point of view in clinical trials, suggest stratifying patients according to the predominant type of fatty liver biopsy in comparison groups (experimental drug vs. placebo).

## Figures and Tables

**Figure 1 ijms-22-06100-f001:**
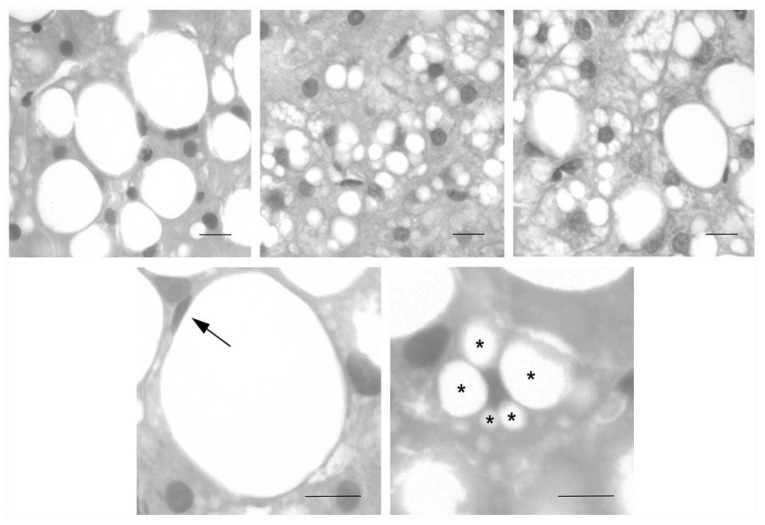
Representative images of liver biopsies performed in the donor regarding macrovesicular steatosis of large (Ld-MaS) and small droplets (Sd-MaS) according to Brunt EM (see [3,4]). In Ld-MaS, a single vacuole of fat larger than half the cell displaces the nucleus to the periphery. In Sd-MaS, the fatty vacuoles are smaller and do not displace the nucleus. Upper panels show hepatocytes with small LD-MaS (left panel) or small Sd-MaS (middle panel) or with both types of macrovesicular steatosis (right panel). The lower panels illustrate a hepatocyte with a single fatty vacuole and a peripheral nucleus (arrow, left panel) and a hepatocyte with multiple fatty vacuoles of different sizes (asterisks) and a centrally located and indented nucleus (right panel). Staining: hematoxylin and eosin. Bars: 20 µm.

**Figure 2 ijms-22-06100-f002:**
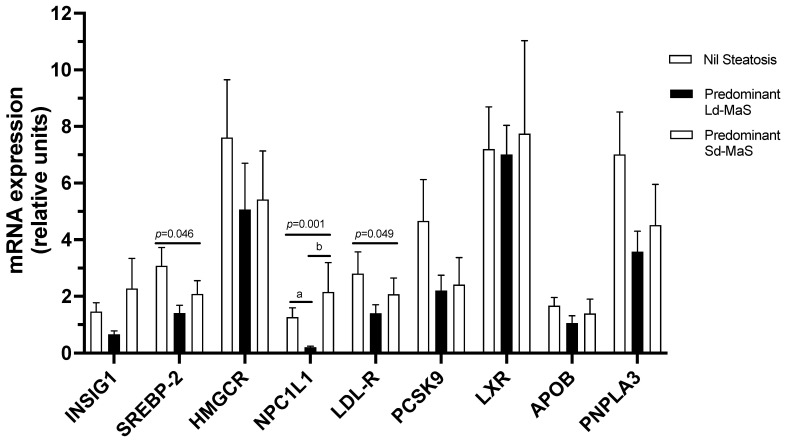
Univariate analyses of liver graft mRNA gene expression of selected proteins involved in lipid metabolism according to the presence and the predominant type of steatosis. The 9 genes are listed on the x-axis, and the fold change of expression is shown on the y-axis. Data are expressed as mean ± SE. *p*-Values refer to overall significance among the 3 groups using the Kruskal–Wallis test; a: *p* < 0.005 at post hoc Dunn analysis with Bonferroni correction for multiple comparisons between predominant Ld-MaS and nil-S groups; b: *p* < 0.02 at post hoc Dunn analysis with Bonferroni correction for multiple comparisons between predominant Ld-MaS and predominant Sd-MaS groups. APOB: apolipoprotein B; PNPLA3: Patatin-like phospholipase domain protein 3; HMGCR: 3-hydroxyl-3-methylglutaryl coenzyme A reductase; INSIG-1: insulin-induced gene 1; LDLR: LDL receptor; LXR: Liver X receptor; NPC1L1: Niemann-Pick C1-Like 1; PCSK9: Proprotein convertase subtilisin/kexin type 9; SREBP-2: sterol regulatory element-binding protein 2.

**Figure 3 ijms-22-06100-f003:**
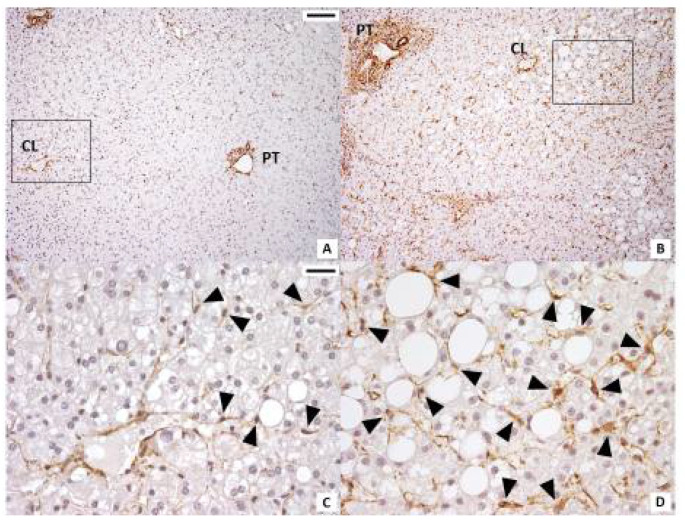
Immunohistochemistry for ⍺-SMA in pre-ischemia liver biopsies. Steatotic hepatocytes accumulate near the centrilobular vein. A larger number of ⍺-SMA-positive hepatic stellate cells (arrowheads) were present near the centrilobular (CL) vein in samples in which Ld-MaS was present inside the hepatocytes. PT: portal tract. (**A**,**C**): liver lobule with Sd-MaS; B,D: liver lobule with Ld-MaS. Original magnification: (**A**,**B**) X100; (**C**,**D**) high-power fields X400. Scale bar: (**A**,**B**) 200 µm; (**C**,**D**) 25 µm. The rectangles in panels A and B delimit the areas shown at higher magnification in panels C and D, respectively.

**Figure 4 ijms-22-06100-f004:**
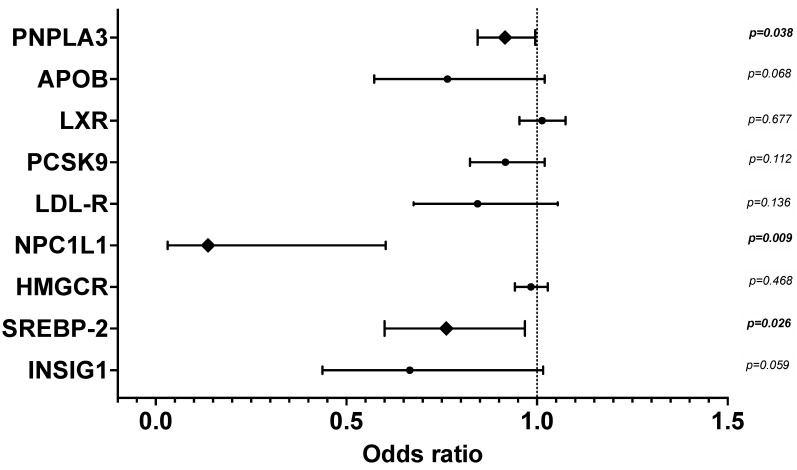
Multivariate binary logistic regression of liver graft mRNA gene expression of selected proteins involved in lipid metabolism, for the presence of predominant Ld-MaS compared to nil-Steatosis. The 9 genes are listed on the y-axis, and the odds ratio with 95% confidence interval is shown on the x-axis. Data were adjusted for donor age, BMI and traumatic cause of death. APOB: apolipoprotein B; PNPLA3: Patatin-like phospholipase domain protein 3; HMGCR: 3-hydroxyl-3-methylglutaryl coenzyme A reductase; INSIG-1: insulin-induced gene 1; LDLR: LDL receptor; LXR: Liver X receptor; NPC1L1: Niemann-Pick C1-Like 1; PCSK9: Proprotein convertase subtilisin/kexin type 9; SREBP-2: sterol regulatory element-binding protein 2. The diamond symbol indicates the significant variables. *p*-values numbers marked in bold indicate numbers that are significant at the *p* < 0.05 level.

**Figure 5 ijms-22-06100-f005:**
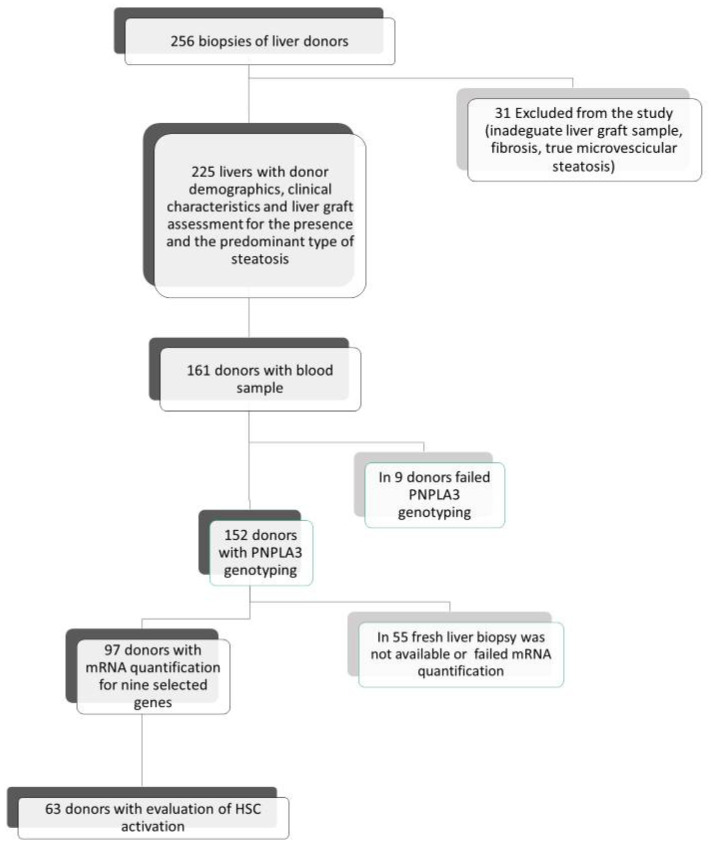
Study population at the time of liver transplantation. Flow chart of the enrolled liver donors according to the presence and the predominant type of steatosis and subgroups analyzed for donor PNPLA3 genotyping, liver graft expression of genes involved in lipid metabolism and activation of HSCs.

**Table 1 ijms-22-06100-t001:** Donor demographics, clinical characteristics, PNPLA3 variant and percentage distribution of Ld-Mas and Sd-Mas calculated both with respect to the total number of hepatocytes and to that of steatotic hepatocytes, according to the presence and the predominant type of steatosis.

	Nil Steatosis	Pred. Sd-MaS	Pred. Ld-MaS	Overall *p*-Value	*p*-Value Pred. Ld-Mas vs. Pred. Sd-Mas	*p*-Value Pred. Ld-Mas vs. Nil Steatosis	*p*-Value Pred. Sd-Mas vs. Nil Steatosis
*n*	74	53	98				
Age (years)	40.50 (23.00–60.25)	50.00 (34.00–62.5)	56.00 (44.00–65.25)	**0.0002**	0.117	**0.0002**	0.393
Males, *n* (%)	43 (58.1)	33 (62.3)	55 (56.1)	0.766	-	**-**	**-**
BMI (kg/m^2^)	24.22 (22.79–25.33)	25.39 (23.23–26.94)	25.76 (24.17–27.76)	**<0.0001**	0.415	**<0.0001**	**0.040**
Obesity, n (%)	1 (1.4)	3 (5.7)	10 (10.2)	0.058	-	-	-
Diabetes, n (%)	5 (6.8)	5 (9.4)	11 (11.2)	0.608	-	-	-
Traumatic cause of death, n (%)	32 (43.2)	18 (34.0)	23 (23.5)	**0.022**	0.166	**0.006**	0.291
Noradrenaline administration, (%)	37 (50.0)	18 (34.0)	60 (61.2)	**0.006**	**0.001**	0.142	0.072
ICU stay (days)	4.00 (2.00–7.75)	3.00 (2.00–4.00)	4.00 (2.00–7.00)	**0.013**	**0.024**	1.000	**0.022**
Serum AST (IU/L)	36.50 (24.00–75.00)	43.00 (27.25–79.50)	35.00 (23.00–59.00)	0.324	-	-	-
Serum ALT (IU/L)	33.00 (15.00–61.50)	32.50 (17.00–52.50)	30.00 (18.00–52.75)	0.797	-	-	-
Liver graft Ld-MaS (percent of total hepatocytes)	-	5.00 (0.00–10.00)	5.00 (5.00–12.25)	-	**<0.0001**	NA	NA
Liver graft Sd-MaS (percent of total hepatocytes)	-	20.00 (7.00–30.00)	0.00 (0.00–3.00)	-	**<0.0001**	NA	NA
Liver graft Ld-MaS (percent of steatotic hepatocytes)	-	20.00 (0.00–33.33)	100.00 (70.00–100.00)	-	**<0.0001**	NA	NA
Donor *PNPLA3*, *n* (%) *				0.053	-	-	-
II	29 (52.7)	25 (73.5)	32 (50.8)				
IM	22 (40.0)	8 (23.5)	20 (31.7)				
MM	4 (7.3)	1 (2.9)	11 (17.5)				

Data are expressed as median and interquartile range or as proportions. Pred. Ld-MaS: predominant large droplet macrovescicular steatosis; Pred. Sd-MaS: predominant small droplet macrovescicular steatosis. BMI: Body Mass Index. ICU: intensive care unit. AST: aspartate aminotransferase. ALT: alanineaminotransferase. PNPLA3: Patatin-like phospholipase domain-containing protein 3. * PNPLA3 genotype and allele frequencies are in Hardy–Weinberg equilibrium (*p* > 0.05). *p*-values numbers marked in bold indicate numbers that are significant at the *p* < 0.05 level.

**Table 2 ijms-22-06100-t002:** Liver graft HSC total and zonal activation according to the presence and the predominant type of steatosis.

	Nil Steatosis	Pred. Sd-MaS	Pred. Ld-MaS	*p*-Value	*p*-Value Pred. Ld-Mas vs. Pred. Sd-Mas	*p*-Value Pred. Ld-Mas vs. Nil Steatosis	*p*-Value Pred. Sd-Mas vs. Nil Steatosis
*n*	22	12	29				
Lobular HSC	4.00 (1.88–7.00)	2.00 (1.00–4.00)	4.50 (3.50–5.75)	**0.041**	**0.036**	>0.999	0.228
Acinar zone 1 HSC	1.00 (0.50–1.00)	0.50 (0.50–1.00)	1.00 (0.50–1.50)	0.204	-	-	-
Acinar zone 2 HSC	2.00 (0.50–3.00)	0.75 (0.00–2.00)	2.00 (1.50–2.50)	**0.041**	**0.035**	0.980	0.303
Acinar zone 3 HSC	1.50 (0.50–2.25)	0.75 (0.50–1.00)	2.00 (1.00–2.00)	**0.043**	**0.037**	>0.999	0.258

Data are expressed as median and interquartile range. Pred. Ld-MaS: predominant large droplet macrovescicular steatosis; Pred. Sd-MaS: predominant small droplet macrovescicular steatosis. HSC: hepatic stellate cell. *p*-values numbers marked in bold indicate numbers that are significant at the *p* < 0.05 level.

**Table 3 ijms-22-06100-t003:** Crude and adjusted linear regression models of variables significantly associated with the percentage of steatotic hepatocytes containing Ld-MaS.

	Crude					Adjusted *				
	B	95% CI		t	*p*	B	95% CI		t	*p*
		Lower	Upper				Lower	Upper		
Noradrenaline administration	0.267	8.095	30.855	3.382	**0.001**	16.562	5.182	28.002	2.874	**0.005**

PNPLA3 polymorphism	0.224	1.303	21.702	2.239	**0.027**	15.284	5.180	25.388	3.006	**0.003**

INSIG1 mRNA	−0.364	−9.157	−1.665	−2.895	**0.005**	−5.102	−9.243	−0.962	−2.473	**0.017**
NPC1L1 mRNA	−0.330	−8.933	−1.138	−2.589	**0.012**	−4.618	−8.695	−0.542	−2.273	**0.027**

Activated total lobular HSCs	0.514	3.934	13.177	3.744	**0.001**	7.766	3.278	12.253	3.510	**0.001**
Activated acinar zone 1 HSCs	0.379	5.337	45.763	2.557	**0.015**	19.668	0.059	39.277	2.034	**0.049**
Activated acinar zone 2 HSCs	0.537	9.635	29.601	3.975	**0.0002**	17.690	7.995	27.385	3.701	**0.001**
Activated acinar zone 3 HSCs	0.463	7.697	32.801	3.263	**0.002**	20.136	8.086	32.185	3.389	**0.002**

* Separate models of multiple linear regression were built for each variable. The noradrenaline administration model was adjusted for donor age and length of ICU stay. All the other models were adjusted for donor age, length of ICU stay and noradrenaline administration. Pred. Ld-MaS: predominant large droplet macrovescicular steatosis; Pred. Sd-MaS: predominant small droplet macrovescicular steatosis. HSC: hepatic stellate cell. *p*-values numbers marked in bold indicate numbers that are significant at the *p* < 0.05 level.

## Data Availability

The data presented in this study are available on request from the corresponding author.

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
