# Peer review of "The Propensity of the Human Liver to Form Large Lipid Droplets Is Associated with PNPLA3 Polymorphism, Reduced INSIG1 and NPC1L1 Expression and Increased Fibrogenetic Capacity"

_ijms, 2021, doi:10.3390/ijms22116100_

Round 1

Reviewer 1 Report

The submitted study is focused on the statistical analysis of factors associated with steatosis. The manuscript is well constructed, scientifically sound and the results are novel and important. What is very important the Authors state the weak points of their study (lines 373-384), which is not often done but at the same time relevant. In my opinion this study deserves to be published after some minor corrections listed below.

In the introduction, more information about the PNPLA3 would be beneficial, especially its role as a therapeutic target.
I would also suggest introducing some information about the association between the PNPLA3 rs738409 variant and other diseases.
Line 41, please remove "3.1. Introduction"
Line 153, it should be "Donor"
Figure 4, does the size of the circles has any meaning? If yes, it should be explained in the caption.
Line 491, I would rather use Lilliefors test. 
The separate "Conclusions" section was not created, instead the conclusison are only briefly summarized (lines 385-393). I would strongly recommend creating such a "Conclusions" section.
Lines 512-513, this should be removed.
Lines 655-658, this should be removed.

Author Response

Reviewer 1

We are grateful for the opportunity to revise our paper ” The Propensity of the Human Liver to form Large Lipid droplets is Associated with PNPLA3 Polymorphism, Reduced INSIG1 and NPC1L1 Expression and Increased Fibrogenetic Capacity” and for the helpful comments.

  • In the introduction, more information about the PNPLA3 would be beneficial, especially its role as a therapeutic target

In the introduction we have added more information about PNPLA3, in particular its role in other diseases and as a therapeutic target ((line 85-126)

  • Line 41, please remove "3.1. Introduction"

"3.1. Introduction" was removed

  • Line 153, it should be "Donor"
    “Donor” was corrected
  • Figure 4, does the size of the circles has any meaning? If yes, it should be explained in the caption.
    We also changed the symbols in Figure 4, indicating the significant variables with rhombus.
  • Line 491, I would rather use Lilliefors test. 

We repeated the tests for normality using the Kolmogorov-Smirnov test with Lilliefors correction but found no changes in the results of the normality tests.

  • The separate "Conclusions" section was not created, instead the conclusison are only briefly summarized (lines 385-393). I would strongly recommend creating such a "Conclusions" section.
    We have created a separate "Conclusions" section as suggested.
  • Lines 512-513, this should be removed.

It was removed

  • Lines 655-658, this should be removed.

It was removed

Reviewer 2 Report

The authors explore the determining factors responsible for large droplet (Ld-MaS) propensity over small droplet macrovesicular steatosis (Sd-MaS) in the liver grafts in this study.  Previously published studies demonstrated the association of liver fibrosis with lipid droplets; however, the tendency to develop liver droplets in donor-liver recipients and the underlying mechanisms are unexplored. The authors conducted a retrospective study on liver biopsies from 225 liver transplant donors and identified no steatosis in 74, predominant Ld-MaS in 98, and Sd-Mas in 53 liver grafts. Authors with linear regression models demonstrated an association of an increase in Ld-MaS in steatotic hepatocytes with a decrease in the hepatic expression of INSIG-1 and NPC1L1, abnormal PNPLA3 M-mediated lipolysis, HSC activation, and progression of post-transplant fibrosis. It is an exciting study to connect an understanding of LD biology and potential fibrotic development in liver grafts.

Lines 170-172: The authors observed more predominant Ld-MaS in liver grafts from older donor age and no steatosis (<5%) in liver grafts from younger donors. However, table 1 shows no difference in the age group of the donors. What is the consideration for the old or young age?
Result 2.2.2.:  Authors show significantly lower NPC1L1 mRNA expression and higher total lobular, acinar zone 2 and 3 HSC activation in liver grafts with predominant Ld-MaS than those with predominant Sd-MaS. It would be of interest if authors could mention the cause and effect relationship. How NPC1L1 M variant, HSC activation interrelated, and Ld-MaS are interlinked, and how are they regulated?

Author Response

We are grateful for the opportunity to revise our paper ” The Propensity of the Human Liver to form Large Lipid droplets is Associated with PNPLA3 Polymorphism, Reduced INSIG1 and NPC1L1 Expression and Increased Fibrogenetic Capacity” and for the helpful comments.

  • Lines 170-172: The authors observed more predominant Ld-MaS in liver grafts from older donor age and no steatosis (<5%) in liver grafts from younger donors. However, table 1 shows no difference in the age group of the donors. What is the consideration for the old or young age?

We corrected also the sentence at line 170-172 because we confirm the data in table 1, there is trend for the donor age but the data doesn’t reach the statistical significance.

  • Result 2.2.2.:  Authors show significantly lower NPC1L1 mRNA expression and higher total lobular, acinar zone 2 and 3 HSC activation in liver grafts with predominant Ld-MaS than those with predominant Sd-MaS. It would be of interest if authors could mention the cause and effect relationship. How NPC1L1 M variant, HSC activation interrelated, and Ld-MaS are interlinked, and how are they regulated?

As suggested, we added in the “Discussion” our hypothesis about the relationship between predominant Ld-MaS, low expression of NPC1L1 and INSIG-1 and activation of HSC.

We can hypothesize that in livers with predominant Ld-MaS, in which the low expression of NPC1L1 and INSIG-1 reflect a high content of free cholesterol, like the hepatocytes, HSC may also have an increased content of free cholesterol and that the latter, as demonstrated in animal models by Tomita et al (Hepatology. 2014), causes a greater activation of HSCs.